# Burnout among Professionals Working in Corrections: A Two Stage Review

**DOI:** 10.3390/ijerph19169954

**Published:** 2022-08-12

**Authors:** Justice Forman-Dolan, Claire Caggiano, Isabelle Anillo, Tom Dean Kennedy

**Affiliations:** Department of Clinical Psychology, College of Psychology, Nova Southeastern University, 3301 College Avenue, Fort Lauderdale, FL 33314, USA

**Keywords:** burnout, occupational stress, organizational stress, correctional staff, interventions, corrections, mindfulness, emotional exhaustion, depersonalization, personal accomplishment, burnout prevention

## Abstract

Burnout is a state of emotional, physical, and mental exhaustion often caused by excessive and prolonged stress. Given the emotionally and often physically demanding nature of the work of correctional professionals, they are at substantial risk of suffering the adverse consequences of burnout. We systematically reviewed (Stage 1) the influence of burnout amongst forensic psychologists, psychiatrists, case workers, nurses, and correction officers. Interventions were then reviewed (Stage 2) at the individual and collective level to examine the effectiveness or efficacy of treatments for burnout among professionals working in corrections.

## 1. Introduction

The development of the concept of burnout has evolved throughout the years, with different views and definitions proposed. Freudenberger (1974) [1], who coined the term burnout, was the first to propose that occupational burnout occurs when an employee becomes psychologically worn out and exhausted from excessive work demands. Although many others have expanded on his interpretation, Maslach (1978) [2] a pioneer in the study of burnout, established that occupational burnout occurs when employees experience “the gradual loss of caring about the people they work with. Over time, they find that they simply cannot sustain the kind of personal care and commitment required in personal encounters that are the essence of their job” (p. 56). Maslach and Jackson (1981) [3] established occupational burnout as a syndrome characterized by emotional exhaustion and cynicism. Their theory outlined three dimensions of burnout, emotional exhaustion, depersonalization, and a reduced sense of accomplishment (Maslach & Jackson, 1981) [3]. They developed The Maslach Burnout Inventory (MBI), which remains the gold standard measure of burnout.

The Maslach Burnout Inventory (MBI) defines emotional exhaustion as feelings of being emotionally overextended and exhausted by one’s work, while depersonalization refers to an impersonal response toward patients. Personal accomplishment refers to feeling successful and competent in one’s performance (Maslach et al., 1996) [4]. The Maslach Burnout Inventory operationalizes two other facets of burnout: cynicism and personal efficacy (Maslach & Leiter, 2016) [5]. Cynicism refers to a negative detachment from work while personal efficacy pertains to personal accomplishment, or lack thereof (Maslach & Leiter, 2016) [6]. Burnout frequently manifests in response to both interpersonal and emotional stressors that are endured in the workplace in accordance with an individual’s ability to cope Heinemann and Heinemann (2017) [7] asserted that burnout transcends feelings of stress, fatigue, and exhaustion in the workplace and can spill over into everyday life.

The association between occupational stress and burnout is complex, encompassing numerous contributing factors. Michie (2002) [8] developed a conceptual model partitioning the causes of occupational stress into five categories based on the content and context of stressors, refining the work of Murphy (1995) [9]. Michie advanced Murphy’s work by examining the relationship between the occupational stressor categories and their impact on the individual, assessing the manifestation of poor physical and mental health as a result. The categories of occupational stress found to be associated with the organizational context of work include: intrinsic to the job, role in the organization, career development, relationship at work, organizational structure, and climate (Michie, 2002) [8]. Stressors intrinsic to the job include long hours, work overload, time pressure, difficult or complex tasks, lack of breaks, lack of variety, and poor physical work conditions. Those involving an individual’s role in the organization consist of unclear work or conflicting roles and boundaries and having responsibility for people. Career development refers to the possibilities for job development, organizations with under promotion, lack of training and job insecurity induce stress. The relationships at work category represents the interpersonal relationship between the individual and management, subordinates, colleagues, and those posing threats to personal safety. Lastly, an organizational culture of poor management and collaborative decision-making, under-compensation, and ineffective communication styles promote stress in the workplace environment. Additionally, this model recognizes that individual differences have a role in one’s risk of experiencing stress. Michie (2002) [9] explains that individuals may be at greater risk of experiencing stress if they lack material and psychological resources, increasing their likelihood of being harmed when exposed to occupational stress. This model also includes the impact of interactions between work and home stress, defined as extra-organizational factors, acknowledging that the negative impact of workplace demands and the demands in an individual’s home life interact with each other and with the subsequent manifestation of burnout.

Burnout is a syndrome impacting individuals across various occupations, and the pervasiveness of burnout research implies widespread concern for its effect on individuals as well or organizations. High prevalence rates of burnout can be seen in occupations that are considered emotionally demanding and people intensive. Occupations routinely displaying high risk of burnout include public safety (police officers, correctional officers, fire fighters, etc.), medical, mental health, education, and social work professionals. Although the roles of individuals in these fields differ significantly, they share high levels of responsibility and risk. Occupational demands of such magnitude create work environments beset with long term exposure to stress and mental health risks.

One unique risk factor for burnout among forensic professionals is the perceived, or actual, threat of violence. Forensic professionals are often subjected to the stressors of verbal and physical aggression which significantly increases their risk of experiencing burnout (Brown et al., 2017) [10]. This stressful environment not only places an individual at higher risk for occupational burnout, but also creates the potential for burnout-related consequences to manifest in one’s personal life. Kanno and Giddings (2017) [11] explored how the length of time a forensic worker spends in direct care of clients with significant pathology is associated with a higher likelihood of displaying significant burnout symptoms. The client–professional relationship contributes significantly to the emergence of stress in these settings. This finding was supported by Johnson and colleagues (2016) [12] who described how working with forensic clients can frequently result in the elicitation of strong emotional responses. Part of forensic professionals’ responsibility includes facing challenging and often aggressive behaviors from their clients. Couple this with the frequently held pessimistic beliefs of treatment efficacy, and it may lead to the development of fixed, negative attitudes which take up a permanent place in forensic professionals’ work ethos (Johnson et al., 2016) [12].

Johnson and colleagues (2016) [12] state, “despite the prevalence of burnout and the associated negative outcomes among professionals working within forensic psychiatric settings, little attention has been directed towards reducing or preventing burnout” (p. 65). While the current literature is limited, a study by Brown and colleagues (2017) [10] reviewed various burnout intervention techniques and found mixed, but generally positive, results for forensic professionals. Lee and Cherniack (2019) [13] suggested that interventions promoting safety, health, and well-being among correction workers are needed. Keinan and Malach-Pines (2007) [14] discussed how the use of relaxation training, cognitive structuring, and stress inoculation training may aid in the reduction of stress levels in correction officers. Additionally, Salyers and colleagues (2015) [15] considered the potential benefits of interventions on stress reduction that target a reduction in emotional labor and exhaustion in probation officers.

The existing literature regarding burnout interventions for forensic professionals working in inpatient and forensic settings includes training (Hill et al., 2010; Norman et al., 2020 [16,17] mindfulness-based interventions (Marconi et al., 2019; Kaplan et al., 2020; Márquez et al., 2021) [18,19,20], psychoeducation (Wampole, & Bressi, 2020; Bagaric & Markanovic, 2021) [21,22], stress management (Ekman, 2015) [23], and workshops (Rollins et la., 2016) [24]. Prior studies assessing burnout intervention programs for correctional staff have concluded that psychoeducational training was effective in reducing feelings of emotional exhaustion, depersonalization, cynicism, and in increasing feelings of general well-being (Hill et al., 2010; Norman et al., 2020; Bagaric et al., 2021) [16,17,22]. Historical findings also suggest that mindfulness-based interventions decrease feelings of depression, worry, anxiety, and perceived stress levels while promoting increased feelings of self-compassion (Marconi et al., 2019; Kaplan et al., 2020) [18,19]. While these results are promising, the studies involved focused solely on one type of correctional workers and did not integrate the various types of forensic professionals that operate in the field. Previous studies also had a narrow population focus, such as looking only at burnout among juvenile justice officers (Ekamn, 2015) [23] or law enforcement officers (Kaplan et al., 2020) [19] within the confines of one justice center, which may limit the generalizability of results. The existing reviews in this area tended to center exclusively on corrections officers, neglecting other correctional professionals. Additionally, these reviews were primarily focused on establishing the presence of burnout and conceptualizing symptoms, foregoing the examination of preventative therapeutic techniques and interventions. The current systematic review builds on these studies by integrating the multifaceted personnel within correctional settings and the subsequent treatment of burnout with the aim of formulating suggestions for effective burnout interventions.

The current study began with a scoping review (to explore any current reviews in this area) followed by a two-stage systematic review identifying (a) any existing literature on burnout among individuals who work in corrections (stage one) and (b) associated interventions for burnout among workers in corrections and forensic psychiatric units, including psychiatric inpatient hospitals and veterans’ affairs organizations (stage two). The first stage explicitly examined burnout amongst forensic psychologists, psychiatrists, case workers, nurses, and correction officers on an individual and group level. The second stage involved the review of interventions developed to temper the negative impact of burnout revealed in stage one. Initial pilot searches to improve the sensitivity of each search term were conducted following the Preferred Reporting Items for Systematic Reviews and Meta-analyses (PRISMA: Moher et al., 2009) [25].

The purpose of this systematic review was to establish a better understanding of how burnout impacts professionals working in correctional settings and to provide a comprehensive review of the interventions used to alleviate the deleterious effects of burnout. This review is also meant to aid administrators in developing or integrating existing preventative measures to combat symptoms of burnout experienced by employees working in corrections.

## 2. Materials & Methods

A scoping search was first conducted to identify any existing reviews in this area using The Cochrane Database of Systematic Reviews and Joanna Briggs Institute (JBI). The Cochrane database subsumes chiefly quantitative research while JBI includes qualitative research, providing a sufficient breadth for the scoping searches. The scoping review examined the existing systematic reviews/meta-analyses in burnout among individuals who work in corrections, including burnout amongst forensic psychologists, psychiatrists, case workers, nurses, and correction officers.

Next, a specific protocol was developed and registered with PROSPERO (CRD42022251793) to avoid duplication and ensure transparency. Studies for stage one of the research were located utilizing PsychInfo, Criminal Justice Database, Sociological Abstracts, PsychArticles, Medline and MARP, Practicums, and Applied Dissertations. Search terms and operators for the primary stage database searches included the following: burnout among forensic mental health professionals, correctional burnout, correctional staff and burnout, burnout and mental health professionals, burnout in forensic settings, burnout in prisons, correctional psychiatric unit and burnout, prison staff and burnout, forensic psychologist and burnout, correctional nurse and burnout, group prison counselor and burnout, prison therapist and burnout, and forensic counselors and burnout. Studies for stage two of the research were located utilizing PsychInfo, Criminal Justice Database, Sociological Abstracts, PsychArticles, Medline and MARP, Practicums, and Applied Dissertations. Search terms and operators for the second stage database searches for interventions included the following: burnout interventions, occupational burnout interventions, burnout prevention, burnout treatment and mental health professionals, reducing burnout in mental health professionals, burnout assessment, mindfulness and burnout, psychosocial interventions and burnout, coping and burnout, prison staff burnout interventions, correctional burnout prevention, resiliency in forensic professionals, and reducing burnout in forensic settings. No timeframe was set for the database searches (given the dearth of studies on this subject).

The Comparator and Outcome (PICO) model (Booth & Fry-Smith, 2004) [26] was used to develop the inclusion and exclusion criteria for individual studies. The PICO model was deemed most appropriate given the assumption that comparison studies would be required within the review to answer the question regarding burnout among various forensic based occupations. References from these initial searches were eliminated if they were duplicates, irrelevant by title, or inaccessible. Remaining references were removed if deemed irrelevant upon a further in-depth evaluation.

Studies for stage one were included if they surveyed burnout amongst forensic psychologists, psychiatrists, case workers, nurses, and correction officers on an individual and group level. Studies were excluded if the observed populations consisted of physicians (non-psychiatric), student trainees, oncology nurses (in non-forensic settings), athletes, therapists working with families (in non-forensic settings), internet mental health service providers, fire fighters, public school teachers, residential childcare staff, nursing home staff members, inmates experiencing burnout, and police department structure reviews. Stage two studies were included if they reviewed or included interventions used to temper the negative impact of burnout among individuals in corrections and psychiatric inpatient settings. Due to the paucity of studies, non-correctional psychiatric inpatient settings were included due to the environmental similarities and parallel presentations of burnout with those found in correctional settings. Exclusion was given to those studies that pertained to the use of therapeutic technique in outpatient settings (e.g., mindfulness used with athletes or during surgery) and studies looking at the relationship between individual personality and burnout symptoms.

## 3. Results

### 3.1. Stage 1: Burnout in Correctional Settings

The systematic review consisted of multiple electronic databases, PsychInfo, Criminal Justice Database, Sociological Abstracts, PsychArticles, Medline and MARP, Practicums, and Applied Dissertations which generated a total of 5528 articles, excluding duplicates. Two raters independently screened all titles and abstracts, rating them based on the adapted checklist (Table 1) and reviewed 125 full-text articles. Of these articles, 114 were excluded due to article type, population, outcome relevancy, and insufficient inferential statistics. Figure 1 outlines the process of inclusion and exclusion.

#### 3.1.1. Inclusion and Exclusion Criteria

The inclusion criteria were based on (a) if the study included a burnout or stress determination, (b) if there was a correlation measure of stress and/or burnout that was organizationally based, (c) if a control or comparison group was included, and (d) if there was a description of how the stressor was correlated with job stress or burnout.

Studies were excluded if they examined a group that did not consist of front-line correctional officers, or a group not employed in an adult correctional facility. Studies were also excluded if the outcomes focused on offender outcomes, prisoner mental health, or prisoner stress. Finally, articles were excluded if they were non-peer reviewed, a book review, an editorial, or a dissertation.

##### Methodological Quality

Five studies met all the inclusion criteria and were subsequently assessed for methodological quality. Two that met all the eligibility criteria received a rating of “excellent”. The remaining three studies met at least 50% of the assessment criteria and received ratings of “good”. There were no studies that received a rating of “fair” due to meeting less than 50% of the assessment criteria. As such, based on the methodological quality criteria displayed in Table 2, all five studies were included.

##### Sources of Occupational Stress

Michie (2002) [8] refined Murphy’s (1995) [9] model of occupational stress categorizing burnout as arising from the context and content of work. This model organizes occupational stressors into five categories: factors unique to the job, role in the organization, career development, interpersonal work relationships, and organizational structure/climate. Stress occurs specifically when a significant conflict exists between the employee and the work demands placed on that employee (Colligan et al., 2006) [32]. Michie (2002) [8] additionally included both individual and extra-organizational outcomes generated from these sources of occupational stress to account for external contributors to burnout. 

##### Characteristics of Included Studies

The articles reviewed varied in location from South Korea, China, France, the United Kingdom, and the United States. The five included studies also varied in their design with samples consisting of correctional officers, secure psychiatric workers, correctional mental health providers, and other unspecified surveillance, security, and support correctional staff. Facilities were all state-run correctional facilities, housing adult offenders. The organizational stressor category, specific occupational stressor, and outcomes identified in each article are outlined in Table 3.

##### Measurement of Burnout

All the reviewed studies, except for one, adopted the definition of burnout proposed by Maslach and Jackson (1981) [3], demonstrated by use of the Maslach Burnout Inventory (MBI). Of the five studies that administered the MBI, half relied on the original English version of the measure. The remaining four studies used validated translated versions of the MBI, including the French (Boudoukha et al., 2013) [27], Korean (Choi et al., 2020) [28], and Chinese (Hu et al., 2015) [31] versions. Further, all studies apart from one, measured all three features of burnout (emotional exhaustion, depersonalization, and personal achievement). Clements & Kinman (2021) [29], administered the abbreviated version of the MBI (Maslach, Jackson, & Leiter, 1996) [33] measuring only emotional exhaustion (i.e., 3 of the 22 items). There are currently five adapted versions of the MBI, developed for the purpose of administering to various groups and settings. One study (Gallavan & Newman 2013) [30] administered the Maslach Burnout Inventory-Human Services Survey (MBI-HSS) designed for professionals in various occupations in the human services field. Another study (Hu et al., 2015) [31] administered the Maslach Burnout Inventory-General Survey, designed for use with professionals employed outside of the human services and education fields.

##### Occupational Stressor Impact

Intrinsic to Job Role

Factors in this category are associated with fundamental work demands, physical and psychological, that impose stress upon employees. Stressors outlined in this category include poor physical working conditions, work overload (over/underload), time pressures (shiftwork/hours), physical danger, autonomy, isolation, and meaningfulness of work. The articles demonstrated that exposure to stressors in this category predicted all three dimensions of burnout, most significantly being associated with emotional exhaustion. Work overload was presented in three of the reviewed articles, likely to be due to the increase in forensic professional’s perceptions of unmanageable workloads as inmate populations continue to rapidly increase despite facilities’ inability to establish sufficient resources to meet such demands (Schiff & Leip, 2019) [34]. The study examining Chinese correctional officers addressed an increased workload, time constraints and working overtime due to the inability to adequately adjust to facility reform policies and staff shortages (Hu et al., (2015) [31]. These require individuals to meet increased physical and psychological demands to maintain functionality of the facility at the expense of personal resources. Another article addressed work overload as a result of role overload, which occurs whenever staff are expected to complete many different tasks in an abbreviated period or without proper resources (Lambert et al., 2020 [35]; Triplett et al., 1996 [36]). Importantly, Choi et al., (2020) [28], identified a significant positive correlation with role overload and the burnout dimension of personal accomplishment, suggesting that when work overload is associated with role conflict/overload, personal accomplishment is not negatively impacted. Overall, workload has direct and indirect impact across stressor categories, influencing negative perception of fair compensation, relationship quality with colleagues and supervisors and overall satisfaction in the workplace. Considering the characteristics of the prison environment and bureaucracy, forensic professionals are in a unique position where, often, they experience high demands and little control over their work (Choi et al., 2020) [28].

Correctional facilities tend to be highly bureaucratic institutions with clients who exhibit negative behaviors, two factors known as correlates of burnout (Ackerley et al., 1988) [37]. All articles reviewed made mention of the correctional environment and the associated risks. As stated in Choi et al. (2020) [28], researchers have begun to examine the relationship between perceived dangerousness of the job and burnout among correctional workers. Several studies indicating perceived dangerousness indicate negative perceptions of the workplace among correctional officers (e.g., Lambert & Hogan, 2010 [38]; Savicki et al., 2003 [39]; Wells et al., 2009 [40]). Despite this issue being addressed in each of the reviewed articles, only one instrumentally assessed the impact of the perceived dangerousness of the work environment. Hu et al. (2015) [31], administered the Work Stress Scale for Correctional Officers to identify the perceived threat in the work environment. Using the MBI-GS, the elevated presence of perceived threat was a predictor of emotional exhaustion, cynicism, and reduced professional efficacy. Hu et al. (2015) [31] further explained that the correctional environment is particularly exposed to an elevated level of psychosocial risk factors, resulting in staff perception of a much more constant threat of danger from those they supervise. The model details the relationship between occupational sources of stress, the individual, and burnout as defined by Michie (2002) [8]. 

Role in the Organization

Stress induced by issues with an individual’s role and responsibility within an organization can be understood in relation to roles with high levels of responsibility and unclear boundaries/demands, resulting in overall role stress encompassing role ambiguity and role conflict (Akanji 2013) [41]. One article assessed role stressors, Choi et al., (2020) [28], finding that officers experience of issues with role clarity was a predictor of depersonalization and lack of personal achievement. Results suggest that officers who are overworked, having many roles within the organization, are significantly less personable toward inmates, exhausted in their work, and lack feelings of accomplishment in the workforce. Additionally, Hu et al., (2015) [31], asserted that not only are correctional officers expected to cope with work demands, but they must also meet societal expectations. Correctional officers have the responsibility of securing inmates, protecting society from them, and preparing said inmates for reintegration into society. Thus, the importance of the reduction of burnout among correctional staff to ensure quality functionality of these facilities is reaffirmed.

Career Development

Explicitly this stressor category addresses the ability to grow and develop within one’s career, represented by factors such as over/under promotion, job security and career development opportunities. Implicitly, this category encompasses overall job satisfaction, representative of the impact stated explicit stressors often have on impeding career progression and negatively influencing an employees’ sense of wellbeing and commitment to work (Akanji, 2013) [41]. Eight items derived from the Job Satisfaction Survey were administered in Choi et al. (2020) [28], written in the form of statements about one’s perceptions toward work. Overall job satisfaction in this study was identified as a predictor of all three features of burnout. Alternatively, Gallavan & Newman (2013) [30] administered the Professional Quality of Life survey, to assess for compassion satisfaction or the feelings of pleasure one derives from doing work well. Compassion satisfaction was associated most strongly with personal achievement. Researchers suggested that this component highlighted the need for a sense of competence, success, and pleasure from one’s work to mediate a decrease in personal achievement. Forensic professionals working in correctional facilities are constantly facing staff shortages, policy adjustments, inmate disputes, and unexpected challenges, creating a work environment providing little time to focus on career development.

Relationships at Work

A significant contributor to occupational stress are challenging workplace relationships, which may involve managers, subordinates, and/or colleagues posing threats of violence, biased opinions, unsupportive management, harassment, dark leadership, artificial social or physical workplace isolation and other deviant behaviors, often causing social disruption (Akanji, 2013) [41]. Forensic professionals are at greater risk of being exposed to verbal, physical, and witnessed violence. Repeated exposure to such potentially traumatic experiences increases the risk for burnout, traumatic stress related disorders, and may increase the risk of experiencing victimization on the job. Five of the reviewed studies made it a point to include the impact of inmate–staff interactions and conflict in their evaluation of occupational stress. Boudoukha et al. (2013) [27] examined the impact of correctional staff exposure to direct (physical, verbal assaults and assaults with weapons) and indirect (witnessed assaults) violence from inmates. Given the significance of frequent repeated exposure to aggressive and violent situations, researchers assessed the presence of PTSD symptoms alongside burnout features. Correctional staff frequently reported experiencing and/or witnessing inmate violence, in turn showing elevated levels of posttraumatic stress symptoms. Strong correlations between posttraumatic stress subscales (intrusion, avoidance, and hyperactivity) and burnout subscales emerged, specifically related to emotional exhaustion and depersonalization. Researchers suggested that “people experiencing both interpersonal violence and stress are particularly prone to experiencing both PTSD and burnout” (Boudoukha et al., 2013) [27]. In fact, elevated levels of emotional exhaustion were associated with increased risk of correctional staff development of post traumatic symptoms.

Similarly, Gallavan & Newman (2013) [30], assessed the presence of traumatic stress symptomology by administering the Professional Quality of Life. This measure includes a secondary traumatic stress subscale designed to assess for “work-related, secondary exposure to traumatic events” (Stamm, 2005, p. 5) [42] and symptoms including reexperiencing the traumatic events, avoidance of reminders, and persistent arousal. The presence of secondary traumatic stress was also a predictor of emotional exhaustion and depersonalization but demonstrated a stronger association with emotional exhaustion. Apart from the traumatic symptomatology component, Choi et al. (2020) [28] examined the impact of victimization (physical and verbal) experiences among correctional officers as it related to the development of burnout features. Again, a strong correlation between emotional exhaustion and depersonalization was found as this relates to officer’s victimization. These results suggest that experiencing verbal victimization may be more impactful on the development of burnout than physical violence, likely due to the persistent nature of verbal attacks versus the isolated nature of physical attacks. Researchers explored the possibility that officers may perceive a greater sense of control over incidents of physical violence as they could protect themselves, whereas verbal violence goes undefended, as it could be seen as unethical and unprofessional for officers to respond. Clements & Kinman (2021) [29], took a similar approach, implementing an aggression tool measuring experiences of verbal abuse and threats, physical assault, and sexual harassment and assault. Results display a direct association between violence and emotional exhaustion. All articles reviewed found workplace aggression, physical and verbal, as significant predictors of emotional exhaustion, while three articles that used the full scale MBI also identified depersonalization as an outcome. Considering the daily relationships with inmates that must be upheld by forensic professionals it is clear how such interactions could result in occupational burnout. Boudoukha et al. (2013) [27] conceptualized that emotional exhaustion in these instances arises because of this pervasiveness of these relationships; subsequently, depersonalization allows employees to avoid inmates, who may become increasingly attention seeking.

Organizational Structure/Climate

Within this category are stressors specific to the stress induced by the organizational structure and climate or daily functionality on employees. Formalization weighs heavy in this category as correctional facilities tend to be highly structured environments creating little opportunity for employee autotomy or collaboration (i.e., ridged management/leadership styles, lack of decision-making participation). Additionally, formalization encompasses inflexible procedures, rules, and regulations impacting employee perception of their work environment (i.e., high work performance expectation, under compensation, working/motivational conditions). Organizational climate was addressed in Hu et al. (2015) [31], finding that correctional officers experienced elevated levels of burnout when they reported exhibiting high effort on the job and receiving low compensation. This can leave a perception of underappreciation in the work environment, as work demands must be met despite lack of reward for fulling required duties. Furthermore, Clements and Kinman (2021) [29] found that correctional staff reported being under compensated when they reported being overloaded with work responsibilities or demands. As higher workload and role responsibility was reported, perceptions of fairness of rewards for efforts decreased, resulting in burnout elevations. Clements and Kinman (2021) [29] also incorporated correctional staff’s relationship with management, as it relates to workload, finding that increases in workload/role overload was associated with decreased perceptions of being treated with dignity and respect by management. These outcomes reveal the importance of recognition and reward for correctional staff effort, as positive reinforcement for meeting heavy job demands may increase productivity and motivation and decrease burnout.

Individual and Extra-Organizational

One study addressed the impact of individual and extra-organizational factors that serve as contributors to the development of burnout among forensic professionals. Individual factors refer to personal characteristics and physiological differences that increase one’s susceptibility to experiencing burnout. Gallavan and Newman (2013) [30], examined optimism through Carver and Scheier’s model (1981), which asserts that optimism influences outcomes by its self-regulatory nature. Unfortunately, correctional mental health professionals in this study demonstrated a negative relationship with optimism, reporting greater negative work experience and reduced ability/motivation to engage and persist. Gallavan and Newman suggested that the lack of perceived control over their environment may induce depleted optimism, in turn decreasing the belief that their actions will impact their circumstances, discouraging them to pursue changes [30].

Extra-organizational factors refer to conflicts in an individual’s work-life balance, indicative of inter-role conflicts that put strain on the ability to meet both work and personal life demands. Gallavan and Newman (2013) [30] explored the most prevalent extra-organizational factor, work–family conflict, addressing an individual’s dual roles in both their work and family, each consisting of separate obligations and duties. Conflict arises when the expectations and demands of one role interfere with another, which can cause pervasive negative impacts on the employees themselves as well as the organization they belong to. Findings indicated that work stress may be substantially related to the intrusion of work demands on family life, contributing significantly to the manifestation of burnout.

### 3.2. Stage 2: Correctional Burnout Interventions

The systematic review consisted of six electronic databases, PsychInfo, Criminal Justice Database, Sociological Abstracts, PsychArticles, Medline and MARP, Practicums, and Applied Dissertations, and generated a total of 32,806 articles, excluding duplicates. Two raters independently screened all titles and abstracts, rating them based on the adapted checklist (Table 4) and reviewed 14 full-text articles. Of these articles, three were excluded due to (a) lack of empirical data, (b) populations in a non-controlled environment, (c) lack of measuring burnout, and (d) no intervention was implemented. Figure 2 outlines the process of inclusion and exclusion.

#### 3.2.1. Methodological Quality

Eight studies met all the inclusion criteria and were subsequently assessed for methodological quality. Five studies that met all the eligibility criteria received a rating of “excellent”. The remaining three studies met at least 50% of the assessment criteria and received ratings of “good”. There were no studies that received a rating of “fair” due to meeting less than 50% of the assessment criteria. As such, based on the methodological quality, all eight studies were included (Table 5).

#### 3.2.2. Assessment of Methodological Quality

An 8-item quality assessment checklist was developed a priori to assess the methodological quality of the included studies, adapted from a previous checklist (Table 4) used by Stergiopoulos et al. (2011) [46]. Assessment questions examined study design, outcome, burnout/stress measures, and population. Methodological quality was independently assessed by raters (Table 5). Studies meeting all assessment criteria were rated as “excellent” while studies that met at least four out of the eight criteria were rated as “good”.

#### 3.2.3. Inclusion and Exclusion Criteria

The inclusion criteria were based on if (a) the study included a burnout or stress determination, (b) there was a correlation measure of stress and/or burnout that was organizationally based, (c) the outcomes had a description of how the stressor was correlated to job stress or burnout, (d) an experimental, quasi-experimental, or non-experimental design was used, and (e) the data analyses were appropriate for the research questions.

Studies were excluded based on the population studied. These included studies examining groups of front-line inpatient and outpatient professionals working in non-psychiatric facilities. Non-forensic inpatient settings were included if they were psychiatric in nature. These studies were included due to what was learned from the scoping review which yielded minimal studies of psychiatric correctional facilities. The highly structured nature and secure environment of inpatient psychiatric facilities closely mimics the conditions found in secure correctional facilities. Studies that did not describe treatment outcomes including mental health, burnout, and/or stress levels or did not include outcomes about the sample population were also excluded. Finally, articles were not included if they were non-peer reviewed, a book review, an editorial, or a dissertation.

#### 3.2.4. Characteristics of Included Studies

The articles reviewed varied in location, with one article from Spain, one from Italy, one from Croatia, one from Sweden, one from the United Kingdom, and the remaining four based in the United States. The eight included studies varied in their design. Further information on sample population, intervention type, and outcomes can be found in Table 6.

### 3.3. Review of Interventions 

Interventions for burnout are present across a multitude of professional fields. However, very little research has focused on interventions for negating burnout among corrections professionals. Of the articles reviewed, three articles focused on psychoeducational workshops or training as a burnout intervention strategy and five assessed mindfulness-based intervention programs. Each program reported significant changes in burnout or psychologically related symptoms.

Mindfulness involves acknowledging present focused feelings moment by moment. Mindfulness based stress reduction programs have been shown to decrease physical and psychological symptoms (Shapiro et al., 2006) [47]. A randomized controlled study of a mindfulness-based stress reduction program found that, post-treatment, individuals receiving treatment had a reduction in saliva cortisol levels, organizational stress, and burnout (Christopher et al., 2018) [48]. Additionally, these interventions were shown to improve alcohol consumption and burnout among police officers (Chiesa & Serretti, 2014) [49]. In nurses, mindfulness-based programs aided in improving psychological distress and stress symptoms along with feelings of overall well-being (Cohen-Katz et al., 2005; Duarte & Pinto-Gouveia, 2016) [50,51]. A study looking at adult and juvenile justice officers found stress reduction-based programs to be highly rated by staff members in terms of the benefits they provided (NIJ, 2000; as cited by Ekman, 2015) [23]. More specifically, justice officers felt that vignettes teaching empathy training, mindfulness skills with breathing exercises, and motivational exercises as most beneficial to their burnout symptoms. Interestingly, the justice officers identified interpersonal communication skills to be of value (Ekman, 2015) [23].

Psychoeducation is an evidence-based therapeutic intervention for patients that provides information and support to better understand and cope with mental illness and corresponding symptoms (Atri & Sharma, 2007) [52]. Several studies demonstrate the value of psychoeducation in prevention and control of mental illnesses ranging from depression to schizophrenia (Atri & Sharma, 2007) [52]. Research regarding psychoeducation and burnout in other non-mental health professions have yielded promising results. Kravitz and colleagues (2010) [53] implemented a single psycho-educational intervention to teach positive self-care behaviors to nurses. Using the MBI to assess burnout symptoms, both emotional exhaustion and depersonalization decreased overall (Kravitz et al., 2010) [53]. Another study conducted by Mustafa (2020) [54] explored to what extent a psychoeducation program affects the burnout levels of university students. These authors reported that the psychoeducation program was effective in reducing the emotional exhaustion and depersonalization scores of the students in the experimental group. These results suggest that psychoeducational workshops and programs may be a good approach to alleviate burnout symptoms in a diverse group of populations.

Of the eight studies included in this review, five assessed the effectiveness of mindfulness-based interventions and the numerous factors associated with burnout including stress, self-compassion, anxiety, depression, and quality of life. These articles assessed mindfulness-based interventions for alleviating burnout and measured changes in mindfulness practices and skills among individuals. The remaining three studies reviewed assessed psychoeducational training and workshop-based interventions implemented in correctional settings and the numerous factors associated with burnout including depersonalization, personal accomplishment, emotional exhaustion, and cynicism. These authors then examined the efficacy of these programs for alleviating burnout and measured changes amongst each individual participant. 

Stress is defined as the presence of external influences that result in tension and strain for the individual, whereas perceived stress is defined as how stressful individuals appraise situations in their lives, which includes the unpredictable, uncontrollable, overload feelings an individual is experiencing (Cohen et al., 1994; Kriakous et al., 2019) [55,56]. Two of the studies examined how mindfulness interventions influenced stress levels. Márquez and colleagues (2021) [20] implemented a seven-week mindfulness-based stress reduction program among 22 national police officers in Palma de Mallorca, Spain. The authors observed a significant difference in the perceived stress scale from pre to post-test reflecting significant improvement. Similar positive results were found by Wampole and Bressi (2020) [21] who examined the effects of a mindfulness-based intervention on inpatient psychiatric nurses. Participants reported that mindfulness positively influenced their ability to “deal with” stressful situations and reported that mindfulness skills were helpful in their lives outside of work. Similarly, Bagaric and Markanovic (2021) [22] assessed a stress-focused mindfulness-based cognitive therapy program among 55 employees across four prisons in Croatia. This intervention consisted of two eight-week mindfulness courses. Post-intervention results indicated that participants found the intervention to be moderately helpful for coping with stress with a significant decrease in stress post-intervention. Marconi and colleagues (2019) found that psychiatric health professionals who completed a compassion-oriented mindfulness-based intervention exhibited significant reductions in anxiety scores on the State-Trait Anxiety Inventory (STAI).

All but one of the examined studies revealed positive results for mindfulness-based programs and reductions in burnout. Bagaric and Markanovic (2020) [22] found a significant decrease in burnout symptoms reported by participants post-intervention; however, these results were not maintained at two month follow up. Marconi et al. (2019) [18] examined an intervention consisting of biweekly classes consisting of compassion-oriented mindfulness-based exercises over the course of 18-weeks among 34 psychiatric health professionals recruited from the G. Salvini Hospital in Milan. The authors found that, post- intervention, there were statistically significant reductions in emotional exhaustion on the Maslach Burnout Inventory (MBI). Wampole and Bressi (2020) [21] found that, for inpatient psychiatric nurses, mean Maslach Burnout Inventory- Human Services Survey (MBI-HSS) scores pre- and post-intervention indicated a minimal decrease in emotional exhaustion, depersonalization, and no change in personal accomplishment. However, the authors noted that results of open-ended questions showed that nurses expressed, prior to the intervention, that burnout influenced their experience of negative feelings, having less tolerance for patients, feelings of frustration, and “pulling away” from patients to conserve energy; whereas, post-intervention, participants reported that the skills learned had a positive influence on burnout and stress-related symptoms regarding their work and personal life (Wampole & Bressi, 2020) [21]. Kaplan and colleagues (2020) [19] assessed the effectiveness of mindfulness-based resilience training (MBRT) with 28 law enforcement officers randomized to an intervention group in order to assess changes in burnout and alcohol use. These authors reported that, post-MBRT, significant changes were observed on Oldenburg Burnout Inventory (OBI) scores. Interestingly, self-compassion (on the OBI) was an important factor significantly contributing to a reduction in burnout. Mindfulness and psychological flexibility did not influence decreasing burnout post-MBRT. Contrary to these positive findings, Márquez and colleagues (2021) [20] found that, post mindfulness-based intervention, scores on the Professional Quality of Life Scale (ProQOL) had no relationship to burnout risk for national police officers.

Mindfulness has been defined as a state of concentration rooted in the present where an individual is aware of their current experience without reacting to it or judging (Meng et al., 2020) [57]. Márquez and colleagues (2021) [20] noted that, for a population of national police officers, mean scores for observing and not-reacting on the Five Facets of Mindfulness Questionnaire (FFMQ) improved from pre- to post-intervention. There was also an increase in the observing and nonreactivity of inner experiences subscales on the FFMQ for psychiatric health professionals (Marconi et al., 2019) [18]. Similarly, Bagaric and Markanovic (2020) [22] found significant changes on the FFMQ observation, non-judgement, and non-response subscales post-intervention; however, changes were not observed in the description and awareness subscales among prison employees. Post MBRT intervention, significant changes were also observed in the Acceptance and Action Questionnaire-II (AAQ-II) and FFMQ-SF among police officers (Kaplan et al., 2020) [19] indicating a change in mindfulness skills for these participants. Additionally, among these individuals, post-intervention results showed that increased mindfulness, but not self-compassion or psychological flexibility, led to a decrease in alcohol use as measured by the PROMIS (v1.0) Alcohol Use-Short Form (PROMIS AU-SF) (Kaplan et al., 2020) [19].

Quality of life as understood by the professional quality of life scale (ProQOL)is comprised of compassion satisfaction, which refers to the satisfaction one derives from completing work tasks well, satisfaction with colleagues, and a feeling that one’s work has broader societal value, whereas compassion fatigue is the experience of working with others that leaves the individual feeling emotional arousal and results in the individual being preoccupied with the trauma experience of their patient (Heritage et al., 2018) [58]. Márquez and colleagues (2021) [20] found that, for national police officers, post-intervention scores on the ProQOL showed increased compassion satisfaction and a non-significant increase on the compassion fatigue subscale. In addition, post-intervention, there was an increase in self-benevolence, mindfulness, and universal humanity with a decrease in over-identification, isolation and self-judgment as measured by the Self-Compassion Scale (SCS). The study by Bagaric and Markanovic (2020) [22] indicated a significant change from pre- to post-intervention and at two month follow up on the Clinical Outcomes in Routine Evaluation (CORE-OM) risk subscale suggesting a change in the appraisal of risk or harm on the job following the intervention. Additionally, these authors reported statistically significant changes on the CORE-OM subscales from pre- to post-intervention indicating a decrease in problems and increased well-being. Marconi and colleagues (2019) [18] specified that, for psychiatric health professionals post-intervention, there were statistically significant reductions in depression scores on the Beck Depression Inventory- II (BDI-II).

Emotional exhaustion is defined as being over-extended and exhausted by one’s work (Hill et al., 2010) [16]. Hill and colleagues (2010) [16] conducted a study on the faculty of an inpatient alcohol dependence facility and found that, before training, staff reported, on average, elevated levels of emotional exhaustion, suggesting that, collectively, the team was highly emotionally exhausted. After the training, MBI emotional exhaustion scores were down slightly, but not to a significant degree. Norman and colleagues (2020) [17] found that the half-day psychoeducational program did not yield a significant decrease in emotional exhaustion. Rollins and colleagues (2016) [24] at the six-month follow-up of the study found a significant decrease in emotional exhaustion when compared to the baseline. These findings indicate that the BREATHE program may help alleviate burnout.

Depersonalization refers to an unfeeling and impersonal response to clients (Hill et al., 2010) [16]. Over half of the participants in the study conducted by Hill and colleagues (2010) [16] were classified in the high range for depersonalization on the MBI prior to training. The psychoeducational training resulted in a decrease of depersonalization scores, but not to a significant degree.

Personal accomplishment measures the extent to which an individual rates their competence and achievement in their work with people (Hill et al., 2010) [16]. Hill and colleagues (2010) [16] found that feelings of personal accomplishment rose significantly, indicating that the psychoeducational training had a positive effect upon team members’ feelings of competence and achievement in their work with clients (Hill et al., 2010) [16]. In contrast, Rollins and colleagues (2016) [24] did not find that the psychoeducational BREATHE program was able to significantly influence respondents’ feelings of personal accomplishment.

Cynicism as a function of burnout is characterized as emotional disengagement from the work situation and feelings of doubt regarding the workplace’s integrity (Abraham, 2000, & Albrecht, 2002; as cited by Mangi & Jalbani, 2013) [59]. It is comparable to the MBI’s depersonalization scale. Norman and colleagues (2020) [17] found a significant reduction level in cynicism following the half-day psychoeducational training. There was a significant main effect, which indicates that after the intervention levels of cynicism decreased over time compared to the control group. Further, this effect was particularly pronounced amongst participants who had taken this job due to practical reasons (Norman et al., 2020) [17]. Similarly, Rollins and colleagues (2016) also reported a decrease in cynicism following the application of the BREATHE program when compared to baseline.

Professional efficacy refers to feelings of competence and successful achievement in one’s work. It is akin to the MBI’s Personal Accomplishment scale. This sense of personal accomplishment emphasizes effectiveness and success in having a beneficial impact on people. Lower scores on this scale correspond to greater feelings of burnout. Norman and colleagues (2020) [17] analyzed the half-day psychoeducational training’s effect on personal efficacy in the place of personal accomplishment. Regarding professional efficacy, a significant intervention effect over time was reported. However, this effect was not sustained in the sensitivity analysis. An additional significant interaction term for age and intervention was demonstrated, where the intervention had a positive effect on professional efficacy with increasing age. This effect did remain significant in the sensitivity analysis (Norman et al., 2020) [17].

### 3.4. Stage 1 & 2 Integration

#### 3.4.1. Specific Populations and Burnout

##### Corrections Staff

Three of the studies evaluated the correlation between burnout and the adverse aspects of the prison environment, specifically occupational stressors intrinsic to the job (i.e., perception of threat) and relationships at work (i.e., interactions with inmates). Boudoukha et al. (2013) [27] screened for the presence of posttraumatic stress symptoms (i.e., intrusion, avoidance, hyperactivity/hyperarousal). Negative symptoms were positively correlated with the burnout dimensions of emotional exhaustion and depersonalization, while personal accomplishment demonstrated a significant negative correlation. Alternatively, Choi et al. (2020) [28] focused on the impact of direct victimizations (i.e., verbal, minor physical, serious physical) perpetrated by inmates, which were all positively correlated with emotional exhaustion and depersonalization. Verbal victimization was the most robust predictor of emotional exhaustion and depersonalization. Similarly, Hu et al. (2015) [31] assessed the perceived threat of danger in the work environment, finding significantly elevated emotional exhaustion and cynicism among those who perceived greater threat of danger. Taken together, these studies identified that emotional exhaustion and depersonalization among this population is heavily exacerbated though the experience of verbal and physical violence, as well as the perception of danger in the environment.

When considering interventions for corrections officers experiencing burnout, one study assessed professionals working in five prisons across Croatia. Results indicated that mindfulness-based cognitive therapy decreased burnout and stress for this population. However, results were not maintained at two month follow up. Nevertheless, problematic feelings decreased, and well-being increased as measured by the CORE-OM. Additionally, a case study by Ekman (2015) [23] evaluated an emotion and mindfulness skill training program. Results of this case study indicated that juvenile justice officers found this intervention program to have positive benefits and reported that they would recommend the program. Specific aspects of the program that officers found beneficial were empathy training, mindfulness skills, and motivational exercises. The officers identified interpersonal communication skills to be especially valuable.

Of the three psychoeducational training interventions, one focused on burnout in corrections officers. Norman and colleagues (2020) [17] reported that there was no intervention effect regarding emotional exhaustion for these correction officers. However, there was a significant decrease in cynicism on the group of correctional officers. This effect was moderated by whether the correctional officer had had taken the job due to practical reasons (benefits, salary, etc.) or for personal reasons (interest/passion for the field). Regarding professional efficacy, a positive effect on the outcome with increasing age was found, suggesting that as an officer progresses through their career they feel a higher sense of personal competence and efficiency (Norman et al., 2020) [17]. 

##### Correctional Mental Health Providers

One study concentrated solely on correctional mental health professionals, independent from the correctional staff team. Gallavan & Newman (2013) [30] recruited professionals from six public medium and maximum-security correctional departments in the United States (Oklahoma, Alabama, Arkansas, Missouri, Wyoming, Pennsylvania). Approximately 72% of the sample reported having acquired their master’s degree, approximately 28% of the sample reported having acquired their doctoral degree, and approximately 52% of the sample reported being licensed. The MBI-HSS was administered to assess for burnout among this population, results displaying positive correlations in the emotional exhaustion and depersonalization subscales and a negative correlation on the personal accomplishment subscale. Researchers suggested that emotional exhaustion may be apparent among individuals with high scores in the negative experience of work component as they “…might be expected to be somewhat detached, fearful, and avoidant of people or circumstances experienced as demanding or needy” (Gallavan & Newman, 2013) [30].

##### Psychiatric Inpatient Facility Professionals

Professionals in psychiatric inpatient facilities were included to augment the data yielded from correctional facilities. The professionals in these settings are exposed to tightly structured facilities with strict protocols and security, similar to those of their forensic counterparts.

Two studies were identified as evaluating mindfulness-based interventions among psychiatric professionals consisting of nurses and psychiatric health providers. Both studies found decreases on the emotional exhaustion subscale of the MBI (Marconi et al., 2019 [18]; Wampole & Bressi, 2020) [21]. Wampole and Bressi (2020) [21] further found decreases in mean scores on the depersonalization subscale with no changes observed in scores of personal accomplishments for psychiatric nurses. Further, Marconi et al. (2019) [18] found a significant decrease in depression and anxiety scores as measured by the Beck Depression Inventory-II and State-Trait Anxiety Inventory respectively for psychiatric health professionals. In terms of mindfulness, Wampole and Bressi (2020) [21] specified that mindfulness skills had a positive influence on the participants’ ability to cope with stressful situations across various settings. Similarly, Marconi et al. (2019) [18] found an increase in the observing and non-reactivity subscales on the FFMQ for psychiatric health professionals.

One of the three identified studies focused on psychoeducational training evaluated inpatient behavioral health workers on an alcohol dependence ward (Hill et al., 2010) [16]. The results of this study suggest that psychoeducational training, when implemented to inpatient staff, has the potential to reduce and alleviate facets of burnout, including depersonalization and emotional exhaustion. Additionally, this type of training was associated with a significant increase in feelings of personal accomplishment (Hill et al., 2010) [16]. The second of the psychoeducational training studies by Rollins and colleagues (2016) [24] observed the changes in burnout symptoms of 109 employees of three U.S. Department of Veterans Affairs (VA) medical centers providing behavioral health inpatient treatment. The inpatient staff underwent the BREATHE program to help alleviate burnout symptoms. The results of the study yielded significant decreases in employees’ emotional exhaustion and feelings of cynicism. However, the participants’ sense of personal accomplishment was not significantly affected by the psychoeducational training (Rollins et al., 2016) [24].

##### Law Enforcement

Two studies exploring the effect of mindfulness interventions on law enforcement personnel were identified. In terms of mindfulness, results from both studies showed significant changes in scores on the FFMQ from pre- to post-intervention (Kaplan et al., 2020; Márquez et al., 2019) [19,20]. Increased mindfulness was found to lower law enforcement alcohol use (Kaplan et al., 2020) [19]. From pre to post-test, mean FFMQ subtest scores on observing and non-reacting improved (Márquez et al., 2021) [20]. Burnout was also affected by the mindfulness interventions. Significant changes were found between pre- and post-interventions scores on the OBI and ProQOL (Kaplan et al., 2020 [19]; Márquez et al., 2021 [20]). Changes in self compassion measured by the SCS were found to have a considerable influence on reducing burnout post-intervention (Kaplan et al., 2020) [19]. SCS results also reflected an increase in self-benevolence, mindfulness, and universal humanity with a decrease in over-identification, isolation, and self-judgment. However, mindfulness and psychological flexibility was not associated with decreasing burnout post-MBRT (Kaplan et al., 2020) [19]. Quality of life scores were found to significantly change post-intervention with a significant increase on the self-compassion subscale of the ProQOL (Marquez et al., 2021) [20]. Finally, stress as measured by the perceived stress scale showed a significant decrease from pre- to post-test measures.

## 4. Conclusions

The results of this review indicate that occupational stressors are significant contributors to the manifestation of burnout among forensic professionals employed within adult correctional facilities. All five organizational factors outlined were identified as influencing burnout dimension elevations, the most impactful being factors intrinsic to the job and relationships at work. Notably, emotional exhaustion presented more frequently and at greater elevations in comparison to the other dimensions of burnout. Given these results, interventions aimed toward decreasing emotional exhaustion and coping with intrinsic and relational factors may be prioritized for implementation within correctional facilities.

Emotional exhaustion was influenced by stressors intrinsic to the job, relationships at work and by the organizational structure and climate of the employer. Four studies were identified assessing the presence of burnout symptoms among correctional staff, encompassing those involved in security, surveillance, and support staff (Boudoukha et al., 2013; Choi et al., 2020; Clements & Kinman, 2021; Hu et al., 2015 [27,28,29,31]. Consistent results were demonstrated across the studies, displaying the presence of all three dimensions of burnout, with emotional exhaustion exhibiting the strongest significance. Each study considered different characteristics of the correctional environment as predictors of burnout among correctional staff. These studies recruited participants from various countries around the world employed in correctional facilities varying in level from medium to maximum. Participants included correctional officers in North-East China (Hu et al.,2015) [31], prison, correctional, and secure psychiatric workers in the United Kingdom, and correctional staff in South Korea (Choi et al., 2020) [28] and France (Boudoukha et al., 2013) [27]. Emotional exhaustion was most elevated when exposed to occupational stressors from each category outlined in Michie’s (2002) [8] conceptual model. Hu et al. (2015) [31] discussed elevations in this dimension when correctional officers reported exhibiting high job effort and receiving low job reward or improper compensation for the effort displayed. Similarly, Clements & Kinman (2021) [29] identified significant elevations in emotional exhaustion when compensation was perceived as unfair. Their study also addressed the deleterious impact of poor communication and treatment from superiors, accompanied by the inability to disclose experiences of stress. Choi et al. (2020) [28] found elevations due to staff being overloaded with work demands, unclear or excessive role responsibilities, and diminished overall job satisfaction. An Internet CBT-based burnout program was found to improve symptoms of disengagement, depression, anxiety, and stress among individuals medically diagnosed with burnout (Zielhorst et al., 2015) [60]. Exercise therapy was found to reduce emotional exhaustion and overall fatigue to a greater extent when compared to a control (de Vries, 2017) [61]. A stress management intervention and group therapy aimed at improving feelings of fairness at the organizational and patient-care level decreased emotional exhaustion among mental health employees and physicians (Paris & Hoge, 2010 [62]; Wiederhold et al., 2018) [63]. In addition, Paris and Hodge (2010) [62] found mental health employee’s reported increased feeling of organizational equity post-intervention. Acceptance and commitment therapy was shown to decrease burnout, stress, anxiety, and depression, and increase well-being, workability, and psychological flexibility among government and health employees in the United Kingdom (Lloyd et al., 2013; Puolakanaho et al., 2020) [64,65]. Emotional exhaustion decreased immediately after MSBR and results held for three months following the intervention among health professionals (Klein et al., 2020) [66]. Ahola et al. (2017) [67] found that cognitive coping training, CBT stress management, and meditative physical exercise interventions decreased emotional exhaustion immediately post-intervention among physiotherapists. A two-day training regarding stress reduction and aggression was found to decrease emotional exhaustion scores as measured by the MBI however, not to a significant degree among staff on an in-patient alcohol ward (Hill et al., 2010) [16]. BREATHE, a daylong workshop for reducing burnout among community behavioral health providers, was found to significantly decrease emotional exhaustion when compared to the baseline at 6-month follow up among behavioral health providers (Rollins et al., 2016) [24].

Depersonalization, also identified as cynicism, was most significantly elevated when exposed to stressors associated with role in the organization and organizational structure/climate among corrections staff. Alternatively, in Hu et al. (2015) [31], this burnout dimension elevated most significantly with those exhibiting high job effort, but findings identified a negative effect with proper compensation, indicating that increased reward would decrease cynicism. Choi et al. (2020) [28] found that unclear or excessive role responsibilities elevated the depersonalization dimension the most significantly. Two of the studies (Choi et al., 2020; Hu et al., 2015) [28,31] identified personal achievement, also recognized as professional efficacy, as exhibiting negative correlations with the organizational stressors as well as other burnout dimensions. Interventions found to be effective in alleviating symptoms include an Internet Cognitive Behavioral Therapy (CBT)-based burnout program implemented with participants medically diagnosed with burnout, which was found to increase participants coping skills and decrease burnout symptoms overall (Zielhorst et al., 2015) [60]. Mindfulness Based Stress Reduction (MSBR) was found to decrease depersonalization symptoms directly after MSBR, at three month and one year follow up among a sample of health professionals (Klein et al., 2020) [66]. Similarly, meditative physical exercise was found to decrease depersonalization immediately post-intervention among physiotherapists (Ahola et al., 2017) [67]. A two-day training regarding stress reduction and aggression was found to decrease depersonalization scores, but not to a significant degree among health professionals (Hill et al., 2010) [16]. Finally, cognitive coping training and a half day psychoeducational training program decreased cynicism among correctional employee and physiotherapist samples (Ahola et al., 2017; Norman et al., 2020) [17,67].

Feelings of personal accomplishment were influenced by an individual’s role in the organization, career development, and relationships at work. MSBR was found to be effective in increasing personal accomplishment immediately post-intervention (Klein et al., 2020) [66]. The two-day training completed by Hill et al. (2010) [16] was found to significantly increase feelings of personal accomplishment. Finally, the review by Norman et al. (2020) [17] found a significant intervention effect over time regarding feelings of professional efficacy suggesting increased feelings of personal efficacy among a sample of correctional employees.

### 4.1. Future Directions

Burnout research may benefit from a shift in focus from assessing the prevalence of burnout, to evaluating its impact on professionals’ ability to function as a team. Identification of how to implement changes in policies and organizational structure in correctional facilities may help make use of the current research findings. Future studies may prioritize what can be done to prevent and control the manifestation of burnout. Additionally, more studies are necessary to better understand the effectiveness and efficacy of burnout interventions for specific types of professionals working in correction settings. Finally, the implementation and evaluation of preventative burnout programs for individuals working in correction are warranted.

### 4.2. Limitations

This two-stage review of burnout among professionals working in corrections was not without its limitations. Much of the research on burnout interventions does not focus on professionals working in correctional settings. Thus, this research expanded its scope to include individuals working in in-patient units to help draw more generalizable conclusions. Further, while there are many international studies that have assessed burnout using various interventions, few focused on diversity considerations among participants. Future research may benefit from considering differences in the effect of burnout interventions for individuals of diverse backgrounds. The American Psychological Association (2006) [6] recommends accounting for individual differences such as gender, gender identity, culture, ethnicity, race, age, family context, religious beliefs, and sexual orientation when implementing evidence-based practices. Additionally, many studies failed to address the long-term effects of burnout interventions, thus it cannot be determined if symptom reduction post-intervention holds for the long term. Future research focusing on the longitudinal effects of interventions is warranted, with a specific focus on preventative measures and diversity issues.

## Figures and Tables

**Figure 1 ijerph-19-09954-f001:**
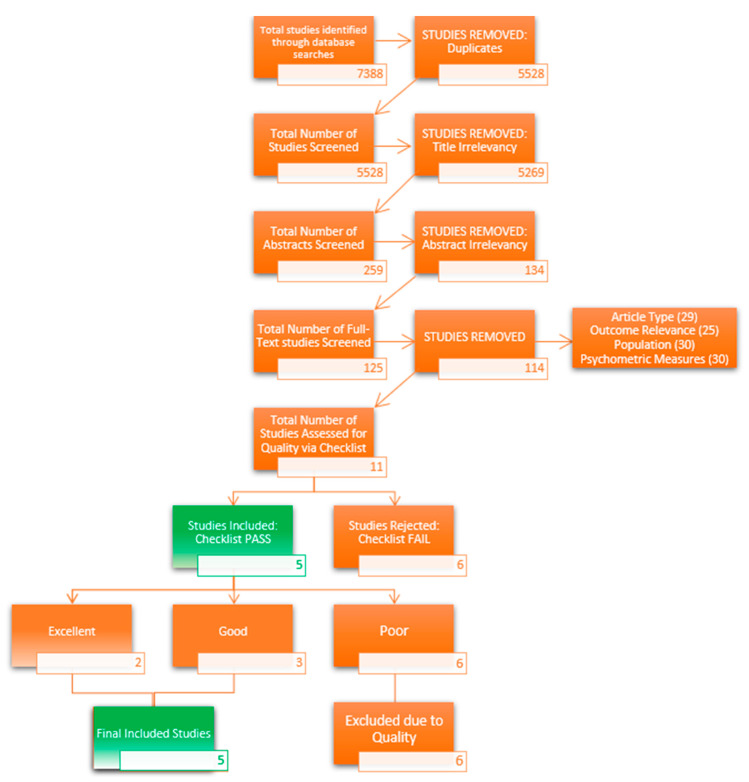
Burnout Systematic Review Flowchart.

**Figure 2 ijerph-19-09954-f002:**
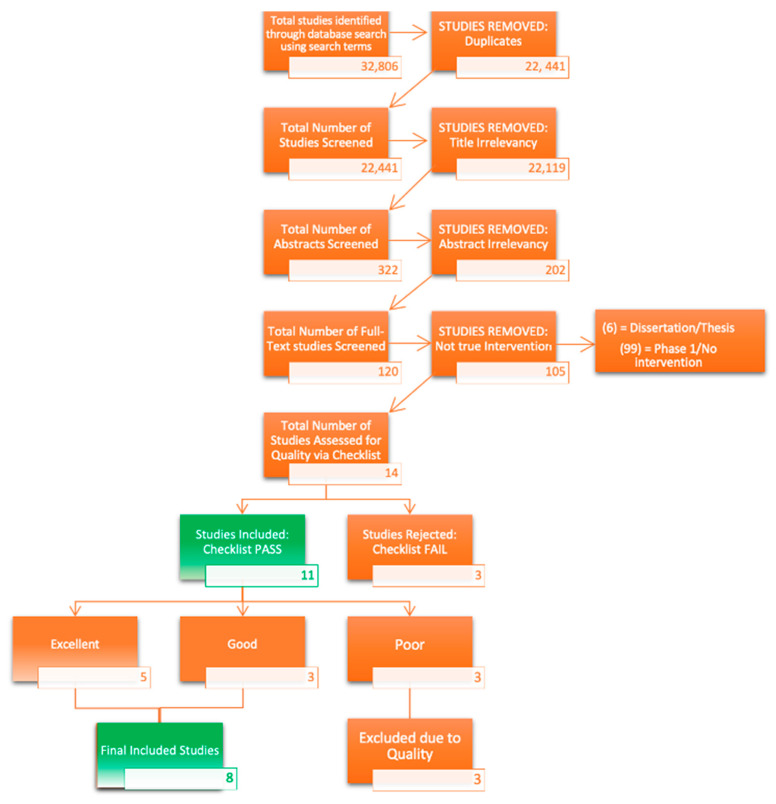
Interventions Systematic Review Flowchart.

**Table 1 ijerph-19-09954-t001:** Stage 1 Quality Appraisal Checklist.

Inclusion Criteria	
**Yes**	
**☐**	**Is the study relevant to the research question**
**☐**	**Diagnosis (one of the following must be checked off a ‘yes’)**
	☐ Burnout (shows symptoms as determined by a valid psychometric measurement and/or biomedical measure)
	☐ Stress (shows symptoms as determined by a valid psychometric measurement and/or biomedical measure)
**☐**	**Correlation (one of the following must be checked off a ‘yes’)**
	☐ Must measure correlates of stress and/or burnout
	☐ Correlates must be organizationally based
**☐**	**Inferential Statistics (both must be checked off as a yes)**
	☐ Includes a control or comparison group
	☐ Were the results directly linked to the aim of the study
**☐**	**Outcomes (must be checked off as ‘yes’)**
	☐ Description of the how the stressor is correlated to job stress or burnout
**Exclusion Criteria**	
**Yes**	
**☐**	**Sample Population (any of the following are grounds for exclusion)**
	☐ A group that does not consist of front-line correctional officers
	☐ A group not employed in an adult correctional facility (i.e., juvenile detention center, juvenile correctional facility, treatment facility, community corrections, probation office, parole office)
**☐**	**No Outcomes (the following are grounds for exclusion)**
	☐ Describes offender outcomes, prisoner mental health, prisoner stress
	☐ No outcomes about the sample population
**☐**	**Type of Article (any of the following are grounds for exclusion)**
	☐ non-peer-reviewed article
	☐ Book review
	☐ Editorial
	☐ Dissertation

**Table 2 ijerph-19-09954-t002:** Stage 1 Quality Assessment Appraisal for Burnout.

Quality Assessment Appraisal					
		Criteria			
Quality of Study	Studies Evaluated	Relevance	Sample Population	Measures and Outcomes	Inferential Statistics
Excellent	Boudoukha et al., 2013 [27] Choi et al., 2020 [28]	Research questions are clear, comprehensive, and clinically sensible.	Selection of samples front-line correctional staff, employed in adult correctional facilities	Burnout and Stressors were measured using clearly defined, reliable, and valid instruments. Outcomes describe correlation between burnout and stressors	Results were directly linked to the aim of the study
Good	Clements & Kinman, 2021 [29] Gallavan & Newman, (2013) [30] Hu et al. (2015) [31]	Some evidence of unclarity, incomprehensiveness or clinical insensibility	Some evidence of unclarity in recruitment	Some evidence of unclarity and lack of psychometric data Outcome lacks complete correlation between burnout and stressors	Some evidence of unclear linkage to study aims

**Table 3 ijerph-19-09954-t003:** Stage 1 Characteristics of included studies.

References	Sample	Burnout/Stress Instrument	Stressors and Organizational Factors	Outcomes
Boudoukha, Altintas, Rusinek, Fantini-Hauwel, & Hautekeete, (2013) [27]	240 CorrectionStaff	Maslach Burnout Inventory—French Version—22-items		Burnout PTSD
		Impact of Event Scale—Revised —22-items	Inmate-to-staff assaults Relationships at work	
		Victimization Index: Inmate-to-Staff Assaults Questionnaire—6-items	Exposure to traumatic event Victimization (direct & indirect) Relationships at work	
		Stress Questionnaire—12-items	Overall stress Relationships at work	
Choi, Kruis, & Kim, Y. (2020) [28]	269 CorrectionOfficers	Maslach Burnout Inventory—Korean Version—22-items		Burnout
		Custody-on-Officer Assaults	Verbal Violence Minor Physical Violence Serious Physical Violence Relationships at work	
		Workplace Factors	Role Clarity Role Overload Role in the organization	
		Job Satisfaction Survey	Perceptions toward Job (8 Items) Career development	
Clements, & Kinman, (2021) [29]	1792 Prison, Correction& Secure Psychiatric Workers	Maslach Burnout Inventory—Abbreviated Version]—3 items		Emotional Exhaustion
		Stress Disclosure—1-item	Ability to discuss stress-related problems with management Relationships at work	
		Aggression Measure	Verbal Abuse/Threats Physical Assaults Sexual Harassment/Assault Relationships at work	
		Organizational Justice Measure—14-items	Workload, Responsibilities, Reward Fairness Supervisors Behavior & Management Organizational structure & climate	
		Health & Safety Executive Management Standards Indicator Tool—8-items	Workload (overload) Working hours Time pressure Incompatible demands Intrinsic to the job	
Gallavan & Newman, (2013) [30]	101 Correction Mental Health Providers	Maslach Burnout Inventory—Human Services Survey—22-items		Burnout
		Professional Quality of Life Survey—Version 5—30-items	Compassion Satisfaction Secondary Traumatic Stress Career development relationships at work	
		Life Orientation Test—Revised	Dispositional Optimism Individual	
		Work-Family Conflict Scale & Family Work Conflict Scale]—10-items	Interrole Conflict Familial Interference with Work Duties Extra-organizational	
		Attitude Toward Prisoners Scale	Attitudes Towards Prisoners Individual	
Hu, Wang, Liu, Wu, Yang, Wang, & Wang (2015) [31]	1769 Correctional Officers	Maslach Burnout Inventory—General Survey—Chinese Version 16-items		Burnout
		Work Conditions—4-items	Work Hours Work Shift Salary Intrinsic to the job	
		Work Stress Scale for Correctional Officers 7-items	Perceived Threat Intrinsic to the job	
		Effort Reward Imbalance 17-items	Job Effort Job Reward Over Commitment Organizational climate/structure	

**Table 4 ijerph-19-09954-t004:** Stage 2 Quality Assessment Checklist.

Inclusion Criteria	
**Yes**	
**☐**	**Diagnosis** (one of the following must be checked off a ‘yes’)
	☐ Burnout (shows symptoms as determined by a valid psychometric measurement and/or biomedical measure)
	☐ Stress (shows symptoms as determined by a valid psychometric measurement and/or biomedical measure)
**☐**	**Correlation** (one of the following must be checked off a ‘yes’)
	☐ Must measure correlates of stress and/or burnout
	☐ Correlates must be organizationally based
**☐**	**Outcome** (must be checked off as a yes)
	☐ Description of the how the stressor is correlated to job stress or burnout
**☐**	**Design (one must be checked off as ‘yes’)**
	□ Experimental
	□ Quasi-experimental
	□ Non-experimental
**☐**	**Data analysis appropriate for the research questions (were the results directly linked to the aim of the study)**
	□ Were the authors interpretations clear
**Exclusion Criteria**	
**Yes**	
**☐**	**Sample Population** (any of the following are grounds for exclusion)
	☐ A group that does not consist of front-line inpatient or outpatient professionals working in a psychiatric or forensic facility
	**No Outcomes** (the following are grounds for exclusion)
**☐**	☐ Does not describe treatment outcomes including mental health, burnout, and/or stress levels
	☐ No outcomes about the sample population
	**Type of Article** (any of the following are grounds for exclusion)
**☐**	☐ non-peer-reviewed article
	☐ Book review
	☐ Editorial
	☐ Dissertation

**Table 5 ijerph-19-09954-t005:** Stage 2 Quality Assessment Appraisal for Interventions.

Quality Assessment Appraisal	
Quality of Study	Studies Evaluated
**Excellent**	Kaplan et al. (2020); Marconi et al. (2019); Márquez et al. (2021); Norman et al (2020); Wampole & Bressi (2020); [17,18,19,20,21]
**Good**	Bagaric & Markanovic (2021); Hill et al. (2010); Rollins et al. (2016) [16,22,24]
**Poor**	White et al. (2015); Eriksson et al. (2018); Suyi et al. (2017) [43,44,45]

**Table 6 ijerph-19-09954-t006:** Stage 2 Characteristics of included studies.

References	Sample	Intervention	Outcomes
Hill et al. (2010) [16]	19 staff on alcohol dependence ward	2-day training	
		Day 1—“Managing stress at the individual, team and organizational level”	Feelings of personal accomplishment increased
		Day 2—“Understanding the causes and consequences of aggression”	Emotional exhaustion and depersonalization decreased slightly
Márquez et al. (2021) [20]	20 National police officers in Spain	7-week mindfulness-based stress reduction intervention	Significant differences in mindfulness, compassion satisfaction, and perceived stress levels
Norman et al. (2020) [17]	166 prison employees across 13 prison wards in Sweden	Group training on everyday conversations	Significantly lowering cynicism
Kaplan et al. (2020) [19]	31 law enforcement officers	8-week mindfulness-based resilience training	Increased mindfulness predicted decreased alcohol use
			Increased self-compassion predicted reduced burnout
Wampole, & Bressi (2020) [21]	8 nurses at psychiatric inpatient unit	12 weekly hour-long psychoeducational sessions based on Dialectical Behavior Therapy’s module on Core Mindfulness	Mindfulness helped participants develop a skill to decrease stress
Marconi et al. (2019) [18]	34 psychiatric health professionals employed at G. Salvini Hospital, Garbagnate Milanese	18-week intervention focused on training mindfulness meditation	Decrease in depression, worry, anxiety, and emotional exhaustion
Rollins et al. (2016) [24]	145 employees across 5 organizations providing behavioral health care services	Burnout Reduction: Enhanced Awareness, Tools, Handouts, and Education (BREATHE)	Small, statistically significant improvement in burnout
Bagaric, & Markanovic (2021) [22]	55 participants, employees of the Ministry of Justice prison system, from four penal institutions: the prisons in Glina, Lepoglava, Gospid, and Zagreb	8 two- hour group workshops	Significantly decreased symptoms 2. Greater experience of subjective well-being and better everyday functioning
			Reduced feelings of stress and burnout

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
