# Peer review of "Burnout among Professionals Working in Corrections: A Two Stage Review"

_ijerph, 2022, doi:10.3390/ijerph19169954_

Round 1
Reviewer 1 Report
I read the review paper on burnout among corrections professionals with great interest. When considering the aim (a review paper), the article has a lot of advantages: the approach to the data is totally professional, the sourced papers seem to be complete, and the description of the method is reasonably sufficient. Yet there are a few errors, which do not permit the article to be published in its current form. I believe however they are easy to fix. I will list them below:
1. The table "inclusion criteria" in the line 200 - it is not described. I also belive it should be changed to a list (and not a copy-paste of the actual tool used in the study)
2. After the abovementioned (no 1) table, the description reads: "Figure 3. Stage 1..." The number cannont be right! How can it be Fig. 3??
3. Then in line 202 we have Figure 5 (??), and in line 203 - Figure 6.
The numbering of tables is a major concern. Also - i would rething if all those tables are needed, as now they bring some confusion.
4. Line 210: (...) Figure 5 all 5 studies were included. Line 205: Twelve studies met all inclusion criteria. So how many it should be? 12 or 5?? WHY?
4. Line 224 - Table 8 (why is the number 8??) - this table needs to be reformated, as now it is not readable. It took me some time to understand: 240 - Correction - al Staff (for example). The headings also read poorly. Needs a major rehaul.
5. Line 295 - Figure 7 - the graph is very nice, but the placing in the text is poor. If the aim was to use it as a road map to the review, it should be specified as such. Now it just pops up in the middle of the text.
6. Line 434 - Figure 1 (sic!) - it should be around the inclusion criteria for part one, not here. Also, it should follow the logic of figure 2 (intervention flowchart), which is good and clear, whereas the burnout flowchart is problematic to read.
7. Line 453 - the table should be changed to a list
8. Figure 9, line 462 - the table is better than in the first part, but nevertheless could be reworked in same style as table for part 1.
9. Line 471 - the description suggests that the autors combined (stacked up) two separate papers (one on causes/corelates of burnout, the other on interventions) due to simmilar content (or other reasons). Nevertheless in a review paper the headline "review of literature" is strongly problematic, especially if it appears in the middle of the paper (unnder the midfulness results). The section headings in part two need thus to be reworked.
10. Relating to my comment 9 - please note that mindfulness reappears as a heaing in line 541...
11. 576. Why separate midfulness intervetions from training intervention?
12. Line 584 - the word "a single" is in different font
13, Line 593 - results heading - Isn't the review a result of review paper?
14. Line 653 - results heading again - I start to be more confused...
15. Line 786 - conclusions - very short. Need to be extended.
16. Line 795 - table for three pages - i do not see the purpose for the table
In general, the paper's content is ok, but it needs an editorial rework to make it readable. Some of the content needs to be summarised and rethought with the general purpose of the article in mind.
Reviewer 2 Report
The authors aimed to review studies on the effects of burnout on professionals working in correctional settings, as well as on effective interventions to alleviate the effects of burnout.
The purpose of the work is clearly defined. Previous research concerning the topic of the article has been sufficiently presented. The results of the review are described in in the text and tables.
However, I would like to draw the authors' attention to several important issues:
- In the introduction part, I recommend to the authors moving the section concerning the relationship between stress and burnout (lines 77-102) after the paragraphs defining occupational burnout (lines 19-46). This will make the introduction structure more precise.
- I would like to suggest that the authors include a paragraph on occupations that are particularly at risk of burnout before moving on to the description of burnout in forensic professionals (before lines 46-47). This will allow a broader view of the multitude of occupations affected by this problem.
- - It would be clearer to the reader if the authors would take care of the correct numbering of tables and figures in the text, in successive order, for example, the authors refer to figure 3 first, and then to 5. I would recommend authors ensure the correct numbering and consistency between entries in the text and the titles of figures and tables.
- I would like to suggest putting Figures 3 and 4 in the appendix.
- I propose the authors shorten some parts in the description of the results and consider them in the discussion part, and then proceed to conclusions.
Round 2
Reviewer 1 Report
I would like to thanks the authors for adressing some of my earlier comments. The text got better, but the tables still need some work on style (prefferded would be a consistent APA style rehaul of all the tables, now they are all over the place).
As for the rest of content it is fine.
Author Response
Tables' fonts, sizing, and general formatting have been corrected to fit APA 7 standards.